# Failure Mechanism of Fiber-Reinforced Prestressed Concrete Containments under Internal Pressure Considering Different Fiber Types

**DOI:** 10.3390/ma16041463

**Published:** 2023-02-09

**Authors:** Zhi Zheng, Ye Sun, Xiaolan Pan, Lianpeng Zhang

**Affiliations:** 1College of Civil Engineering, Taiyuan University of Technology, Taiyuan 030024, China; 2State Key Laboratory of Mechanical Behavior and System Safety of Traffic Engineering Structures, Shijiazhuang Tiedao University, Shijiazhuang 050043, China

**Keywords:** PCCV, FRC, internal pressure, failure mechanism

## Abstract

Current investigations of performance improvement in prestressed concrete containment vessels (PCCVs) with fiber reinforcement are scarce, and the type of fiber to select for PCCVs is not explicitly stated. The failure mechanism of PCCVs with fiber reinforcement under internal pressure is investigated in this paper. The effects of different fiber types, including rigid fiber, flexible fiber, and hybrid fiber, are considered for the creation of fiber-reinforced PCCVs. The mechanical behavior between conventional and fiber-reinforced PCCVs is scientifically compared and identified. The results demonstrate that to achieve the aim of inhibiting early cracking of the concrete, any type of fiber can be taken into account. The performance of the ultimate pressure capacity and yielding of the liner can be promoted, respectively, by introducing steel, steel-PP, and steel-PVA fiber-reinforced concrete. Additionally, the failure regions can be controlled to a certain extent under ultimate internal pressure via the appropriate use of FRC.

## 1. Introduction

Since the severe accident that triggered meltdowns and explosions at the Fukushima Daiichi nuclear plant following the 2011 Great East Japan Earthquake, continuously strengthening the safety and reliability of important nuclear power structures has become an urgent scientific problem to be solved. A containment structure will be severely damaged if affected by accidents that go beyond its design basis and if it lacks enough ductility and energy dissipation capacity. As the key component for the safety of a nuclear power plant, a containment structure should be designed to prevent the leakage of radioactivity material in all cases. To calibrate the margin of safety of the containment under devastating earthquakes, the pioneering work of seismic proving tests for RCCVs can be traced to the end of the 20th century, and was conducted by Taira et al. [1] and Nakamura et al. [2,3]. After their work, the RCCV model was further systematically tested by Hirama et al. [4,5,6]. They found that all maximum seismic responses in terms of shear deformation angle, shear stress, and rebar stress were less than the allowable value, meeting the structural integrity standards under the designed seismic load. Additionally, several investigations have been conducted to research the failure mechanism of PCCVs. By establishing a model of a PCCV at a 1:4 scale, the Sandia National Laboratories performed a string of beyond-design internal pressure tests [7]. By summarizing the results of the limit state and structural load capacity tests, Hessheimer et al. [8] demonstrated that, followed by concrete cracking, steel liner yielding and tearing appear, resulting in functional failure of the PCCV. Parmar et al. [9] carried out ultimate load capacity assessment of a 1:4 scale BARC containment test model, and the results in terms of displacement, stress, and failure mode were obtained.

Except for the above tests and with the development of numerical simulation technology, numerical simulations concerning the structural failure mechanisms of the containment have also been conducted in many studies. By establishing a finite element model that considers material and geometry nonlinearity, Hu and Liang [10] predicted the ultimate bearing capacity and the failure mechanism of a reinforced concrete containment located at Kuosheng Nuclear Power Plant under internal pressure. To verify their modeling method for the PCCV, a 1:4 scale simplified finite element model that considers the unbounded tendons and the axisymmetric characteristic of the model was built by Kenji et al. [11]. Comparing the mechanical behavior of the containment model with test records, Ghavamian et al. [12] performed an internal pressure assessment of the established model and consequently predicted the potential failure mode. Kwak et al. [13] performed numerical simulation analyses using two sorts of developed concrete constitutive models. Their effectiveness and practicability were also calibrated. The focus of the paper by Patrick [14] was the internal pressure response of a steel liner. It was observed that the steel liner in the vicinity of holes and concrete was very easy to detach. Zhang et al. [15] constructed a representative model of a PCCV using the reliable nonlinear mechanical behavior of materials. The safety margin and performance of the model were scientifically studied under internal pressure. Yan et al. [16] also predicted the potential failure mechanisms of PCCVs subjected to internal pressure using a very refined finite element model.

Although the importance of PCCVs in preventing the leakage of radioactive materials is non-negligible, relatively few attempts in recent decades have been made to improve crack resistance and ultimate capacity under internal pressure, which could provide valuable time for the disaster rescue of nuclear power engineering systems and the rapid evacuation of human beings. A wide review of the literature demonstrated that structural responses can be dramatically improved by using fiber-reinforced concrete in important regions [17,18]. In addition, the residual compressive strength and spalling resistance of cement mortar can be enhanced when it is exposed to elevated temperatures by adding fibers into the cementitious composites [19,20]. As a result, the application of FRC in the construction of new structures or repair of damaged infrastructures in recent decades has gradually increased [21,22,23,24,25,26,27]. Specifically, the only work conducted by [28] is worth noting, in which an internal pressure performance evaluation of PCCVs with two types of fiber reinforcement was performed. However, owing to the limited fiber types used and insufficient analysis, the failure mechanism of the containment subjected to the internal pressure with fiber reinforcement has not been systematically investigated, particularly for the crack evolution of concrete. Meanwhile, previous investigations do not explicitly state which type of fiber can be applied to the containment, which greatly hinders the applicability of FRC to the containment. Additionally, apart from the above investigation, specific methods for the design of a containment with fiber reinforcement have not been introduced for current codes. Hence, a comprehensive investigation is required and would be indispensable in thoroughly grasping the failure mechanism of a fiber-reinforced containment. In order to overcome the difficulty of the tests, including the scale of the prototype and the selection of the type of fiber, an alternative way to investigate the mechanical behavior and failure mechanism of fiber-reinforced containments is to choose the finite element simulation method. This study investigates the nonlinear response of an elaborate containment with increasing internal pressure using ABAQUS software and considers the same containment with up to six types of fiber: basalt fiber, carbon fiber, polypropylene (PP) fiber, steel fiber, steel–polypropylene (steel-PP) hybrid fiber, and steel–polyvinyl alcohol (steel-PVA) hybrid fiber. The effect of internal pressure on the failure mechanisms and deformation responses of containments with and without fiber strengthening are evaluated, which ccould facilitate the design of containment while taking into account fiber reinforcement.

## 2. Finite Element Model

### 2.1. The Geometry of the Containment

Referring to China’s advanced nuclear power project, as displayed in Figure 1, the containment model is intended to scientifically simulate the mechanical behavior of the prototype. The finite element model in this study is composed of a hemispherical dome, an upright cylinder, a sealed steel liner, several penetrations, buttress columns, and a prestressed tendon system. The total height of the containment reaches 69 m and the diameter of the hemispherical dome stands at 20 m. The thicknesses of the cylinder, dome, and steel liner are 1.1 m, 1 m, and 6 mm, respectively. Among the penetrations in the cylinder, only the largest hole is considered, which is 7.0 m in diameter. To avoid early failure and stress concentration of the position near the hole, a local thickening area is explicitly provided, and the size is shown in Figure 1b.

The prestressing tendon system consists of horizontal and inverted “U” tendons which are prestressed using the post-tensioning method (see Figure 2). A total of 190 Horizontal tendons are anchored and tensioned at two symmetrical buttresses, while the 140 inverted “U” tendons are constrained at the base slab. In addition, two layers of reinforcement are taken into consideration, as well.

### 2.2. Analytical Model and Finite Element Mesh

The finite element models of the containment structure were built using ABAQUS code. The concrete was modeled using eight-node solid elements with a reduced integration scheme (C3D8R). The prestressing tendons were modeled using the truss element T3D2. Compact and uniform reinforcements are represented by the surface element SFM3D4, ignoring the relative slippage between the steel material and the concrete. Reduced integrated 4-noded general shell elements (S4R) were used to simulate the steel liner. Taking the accuracy into consideration, convergence analysis was performed for mesh sizes ranging from 0.6 m to 0.9 m with an equal difference of 0.1 m. Figure 3 describes the vertical displacement of point A (at the top of the dome) and the radial displacement of point B (at the apex of the hole) of the containment versus tensioning stress with different mesh sizes. It is found that the element sizes equal to or less than 0.8 m have almost consistent results, while the results for element sizes greater than 0.8 m (i.e., 0.9 m) diverge from those in stages with prestressing conditions (Figure 3b). Moreover, the computational efficiency of these different models is also compared to that before applying the internal pressure. The results show that the calculation time is 365 min for a mesh size of 0.6 m, 202 min for 0.7 m, 15 min for 0.8 m, and 12 min for 0.9 m (CPU: Intel i7-7700HQ, memory: 16G DDR4, threads: 8), respectively. Given all of this, finite element analysis was this study is conducted with a coarse element size of 0.8 m

### 2.3. Material Properties and Constitution

#### 2.3.1. Plain Concrete

The strength grade of C50 plain concrete was modeled using the concrete damaged plasticity (CDP) model provided by ABAQUS software, which was first proposed by Lubliner et al. [29]. Table 1 illustrates the material property parameters. Referring to the Chinese code (GB50010-2002) [30], stress–strain curves in uniaxial compression and tension were adopted in the numerical study and are described in Equations (1), (2), (5), and (6) and Figure 4. Meanwhile, two variables ($d_{c}$ and $d_{t}$) in the CDP model provide a scientific description of the damage mechanism under the energy equivalence assumption [31] and are described as Equations (3) and (7) [32]. Inelastic or crack strain (${\tilde{\varepsilon }}_{c}^{\mathrm{in}}$ or ${\tilde{\varepsilon }}_{t}^{\mathrm{ck}}$) and the equivalent plastic strain ${\tilde{\varepsilon }}_{c}^{\mathrm{pl}}$ were required for the CDP input parameter and were obtained by transforming from the constitutive curve using Equations (4) and (8).
(1)$$\begin{array}{c} \sigma_{c}= \end{array}\{\begin{array}{c} \alpha_{a}x_{c}+\left(3-2\alpha_{a}\right)x_{c}{}^{2}+\left(\alpha_{a}-2\right)x_{c}{}^{3}\begin{array}{cc} , & x_{c}\leq 1 \end{array}\\ \frac{x_{c}}{\alpha_{d}{\left(x_{c}-1\right)}^{2}+x_{c}}\begin{array}{cc} , & x_{c}>1 \end{array} \end{array}$$
(2)$$x_{c}=\frac{\varepsilon_{c}}{\varepsilon_{c,r}}$$
(3)$$d_{c}=1-\sqrt{\frac{\sigma_{c}}{E_{0}\varepsilon_{c}}}$$
(4)$$\{\begin{array}{c} {\tilde{\varepsilon }}_{c}^{\mathrm{in}}=\varepsilon_{c}-\frac{\sigma_{c}}{E_{0}}\\ {\tilde{\varepsilon }}_{c}^{\mathrm{pl}}={\tilde{\varepsilon }}_{c}^{\mathrm{in}}-\frac{d_{c}}{1-d_{c}}\frac{\sigma_{c}}{E_{0}} \end{array}$$
(5)$$\begin{array}{c} \sigma_{t}= \end{array}\{\begin{array}{c} 1.2x_{t}-0.2x_{t}{}^{6}\begin{array}{cc} , & x_{t}\leq 1 \end{array}\\ \frac{x_{t}}{\alpha_{t}{\left(x_{t}-1\right)}^{1.7}+x_{t}}\begin{array}{cc} , & x_{t}>1 \end{array} \end{array}$$
(6)$$x_{t}=\frac{\varepsilon_{t}}{\varepsilon_{t,r}}$$
(7)$$d_{t}=1-\sqrt{\frac{\sigma_{t}}{E_{0}\varepsilon_{t}}}$$
(8)$$\{\begin{array}{c} {\tilde{\varepsilon }}_{t}^{\mathrm{ck}}=\varepsilon_{t}-\frac{\sigma_{t}}{E_{0}}\\ {\tilde{\varepsilon }}_{t}^{\mathrm{pl}}={\tilde{\varepsilon }}_{t}^{\mathrm{ck}}-\frac{d_{t}}{1-d_{t}}\frac{\sigma_{t}}{E_{0}} \end{array}$$

Apart from the hardening/softening law, the extra plasticity conditions also need to be considered. The values of related the parameters are shown in Table 2.

#### 2.3.2. Modeling of FRC

In general, fiber added to a concrete mix is mainly classified as one of three types: stiff fiber, flexible fiber, or hybrid fiber combining the characteristics of the stiff fiber and flexible fiber. As different fibers have different characteristics, and thus, the mechanical behavior of FRC may differ, up to six types of fiber, including basalt fiber, carbon fiber, PP fiber, steel fiber, steel-PP fiber, and steel-PVA fiber, were used to investigate their effects on PCCVs. To make full use of the performance improvement for each type of fiber, the optimum characteristic parameters of fibers that play a key role in improving concrete mechanism properties and achieving high performance were adopted with the help of theory and test data from the literature [33,34,35,36,37,38,39,40,41,42,43,44]. The properties of the selected fibers are shown in Table 3. For the practical applicability of the selected fibers, the mix design of the FRC should be equivalent to the concrete mix design of the containment. The corresponding material parameters for the mix design are provided per cubic meter: ① 347.5 kg of P.O 42.5 ordinary Portland cement; ② 664.5 kg of fine aggregate (sand with a fineness modulus of 2.3~2.5); ③ 1181.3 kg of coarse aggregate (graded continuous gravel with a particle size of 5 mm∼20 mm and clay content <1%); and ④ 72 kg of water.

Figure 5 shows plots of the stress–strain curves for each FRC. As is shown in Figure 5a, through the inclusion of fibers into concrete, the peak tensile stress of concrete is strengthened to varying degrees, wherein the steel and steel-PVA fibers achieve the two greatest increases in peak tensile stress, reaching 3.75 MPa and 3.55 MPa, respectively. It should be emphasized that although the peak tensile stress is less increased by the inclusion of steel-PVA hybrid fiber, the post-peak stress for the steel-PVA FRC declines most slowly. As can be seen in Figure 5b, in comparison with the plain concrete, the peak compressive stress for basalt, steel-PP, and steel FRC is increased by 11%, 27.8%, and 62.3%, respectively, while these values for carbon, PP, and steel-PVA FRC are reduced by 5.4%, 0.3%, and 16.7%.

Due to inadequate and insufficient studies on FRC in the aspect of yield criteria and plastic potential, the values of the parameters $\sigma_{b0}/\sigma_{c0}$ and $K_{c}$ for yield criteria and ψ for plastic potential applied to plain concrete were used to analyze the fiber-reinforced concrete. Meanwhile, to effectively identify and predict the damage degradation of the the FRC material, the damage model based on the energy equivalence assumption and fully thermodynamic equivalence applies equally well to FRC, as shown in Equations (3) and (7). This assumption yields almost identical results, with modification of the parameters $\sigma_{b0}/\sigma_{c0}$ from 1.16 to 1.6, $K_{c}$ from 0.667 to 0.7, and ψ from 15° to 36°, which can be attributed to the fact that the PCCV mainly bears tensile stress rather than compressive stress under internal pressure and that the principal tensile stresses in two or three directions are independent of each other.

#### 2.3.3. The Material Property Parameters of Steel

According to the Chinese standard (GB50017-2017) [45], the stress–strain relationship of steel materials was determined and an elastic perfectly plastic model was assumed for the steel liner, reinforcements, and prestressed tendons. Table 4 provides the material property parameters of the steel material.

## 3. The Failure Mechanism Analysis for the Conventional PCCV

To accurately simulate the real service state of the structure under internal pressure, three steps needed to be applied successively, including gravity load, prestressing forces, and internal increment pressure. The first step was to apply the deadweight of the whole structure. For the next step, the effective prestressing force Δσ (control stress of $0.8f_{\mathrm{ptk}}$) was exerted, adopting the decreasing temperature means, and the formula is shown in Equation (9). Finally, uniform increment pressure was applied to the internal surface of the containment until the structure failed.
(9)$$\Delta T=\frac{\Delta \sigma }{\alpha E_{p}}$$
where ΔT is the difference in cooling temperature; Δσ presents the control stress of $0.8f_{\mathrm{ptk}}$; α and $E_{p}$ are, respectively, the thermal expansion coefficient and Young’s modulus of the prestressing tendons.

### 3.1. Deformation Response for the Conventional Containment

To manifest the whole process of the containment from elastic to plastic, load-deflection characteristic and deformation profiles are first investigated in this section. Figure 6, Figure 7 and Figure 8 describe the radial displacement of point A near the edge of the equipment hatch hole, the vertical displacement of point B at the apex of the dome, and the relationship between the displacement contour of the containment and gradually increasing pressure at the mean line of the equipment hole, respectively. Reaching the ultimate internal pressure, the radial displacement of point A and the vertical displacement of B reach 9.3 cm and 11.2 cm, respectively. With the increase in the inner pressure, the containment expands with obvious outward deformation, while the region in the vicinity of the hole contracts visibly and inwards.

### 3.2. Pressure Performance of the Steel Liner

The steel liner remains elastic up to the design-basis internal pressure (0.4 MPa), as shown in Figure 9. With the continuation of inner pressure increasing to 0.95 MPa, the steel liner becomes plastic and begins to yield, with the principal tensile strain exceeding the value of 0.0016, and after that, steel liner tearing may appear. As a consequence, an uncontrolled increase in the leakage rate of the containment may have occurred. Equivalent plastic strain represents the accumulation of plastic strain in the process of structural deformation, and the corresponding value of the steel liner subjected to maximum internal pressure is also included in Figure 9. It is noted that the regions at the head and foot of the equipment hatch hole have obvious plastic deformation. Additionally, plasticity tends to develop at the places in proximity to the thickening area near the equipment hatch wherein stress is concentrated due to the discontinuity of the thickness of the containment cylinder.

### 3.3. Pressure Performance of Prestressing Tendons

With the completion of the prestressed tension and the beginning of the internal pressure loading, the maximum principal stress for the total prestressed tendons reaches 1488 MPa (control stress of $0.8f_{\mathrm{ptk}}$), as presented in Figure 10. When the prestressing tendons begin to yield, the internal pressure is 1.57 MPa for horizontal tendons and 1.64 MPa for “U” type tendons. It should be noted that the ultimate internal pressure is determined as the ultimate strength that is achieved by all prestressing tendons. Further observing the failure contour in Figure 10, the increasing pressure results in the horizontal prestressing tendons reaching the ultimate strength at both the top and bottom of the equipment hole and the “U” type prestressing tendons across almost the whole dome and under the equipment hatch.

### 3.4. Evolution Rule of Cracks and Principal Tensile Strain in Concrete

The evolution rule of cracks and principal tensile strain in concrete are displayed in Figure 11. It can be found that the concrete strain is increased with the increase in internal pressure. As soon as the internal pressure reaches 0.585 MPa, the maximum principal tensile strain value of the containment rises to 0.00015, exceeding the peak tensile strain of the concrete. This means that the containment changes from an almost compressive state to a partially tensile state at the value of pressure. Cracks first arise from the vicinity of penetration and the cylinder’s bottom area. When the internal pressure reaches 0.8 MPa, a small number of cracks develop around the thickening area in the oblique direction. As it rises to 1.1 MPa, some skew cracks and one clear horizontal crack emerge at the hole, the base of the dome, and the base of the cylinder, respectively. With the internal pressure reaching 1.4 MPa, a great number of horizontal cracks develop. Until the pressure exceeds the ultimate internal pressure, the cracks expand significantly in the whole structure, indicating that the containment is completely damaged. Based on the above-mentioned analysis, it is concluded that the structure exists invariably in a secure condition and maintains a certain margin of security under the design internal pressure. For the beyond-design internal pressure, accompanied by the full development of a concrete crack, the steel liner and prestressing tendons consecutively yield, resulting in the containment ultimately collapsing.

## 4. The Failure Analysis for the FRC Containment under Internal Pressure

The consistent load path adopted for the conventional containment was also employed for the fiber-reinforced PCCV to compare the failure mechanism between the conventional containment and the containment with fiber reinforcement under internal pressure. The design-basis internal pressure of the PCCV was 0.4 MPa.

### 4.1. Deformation Response and Failure Modes for the Fiber-Reinforced PCCV

Figure 12 illustrates the corresponding relationship between the radial displacement of point A and the increasing internal pressure while regarding the vertical displacement of point B; the relationship is presented in Figure 13. To make a clear comparison of the displacement between the conventional and fiber-reinforced containment models, points A and B for the containment with fiber reinforcement were selected at the same location of the conventional containment. It is seen that upon inclusion of steel fiber, steel-PP fiber, and steel-PVA fiber, the degradation of the structural stiffness is substantially lower. It is also observed that the addition of basalt fiber, carbon fiber, and PP fiber appears to have a minor impact on the deformation capability of the containment. Specifically, the vertical displacement for point B is 9.97 cm, 10.77 cm, and 9.84 cm for steel fiber, steel-PP fiber, and steel-PVA fiber, respectively, with the value reduced by 10.2%, 3.8%, and 12% compared to the conventional containment under the maximum internal pressure. In the same measure, the value of displacement for point A reduces from 9.3 cm to 7.1 cm, 7.9 cm, and 5.5 cm for containments with steel fiber, steel-PP fiber, and steel-PVA fiber in the radial direction, respectively, with reductions of 23.7%, 15.1%, and 40.9%. Figure 14 charts the deformation profile at the centerline of the hole for fiber-reinforced PCCVs. There seem to be some similarities in failure modes between conventional and fiber-reinforced PCCVs as both containments expand with obvious outward deformation, while the region in the vicinity of the preformed hole contracts visibly and inwards. However, a remarkable decrease in inward contraction near the hole occurs for the steel-, steel-PP-, and steel-PVA-reinforced containments. The performance metrics of the different containments are shown in Table 5.

### 4.2. Pressure Performance of Steel Liner

Figure 15 shows that the principal tensile strain of the steel liner can be divided into two stages. Before reaching 1.1 MPa, the steel liners in any kind of containment present the same characteristics because the steel liner maintains elasticity. After that, the principal strain of the steel liner for both types of containment will increase sharply, including the conventional containment and the containment with the addition of basalt fiber, carbon fiber, and PP fiber. However, the increasing tendency for steel fiber-, steel-PP fiber-, and steel-PVA fiber-reinforced containments will slow down. Figure 15 also presents the equivalent plastic strain of the steel liner as the internal pressure reaches the maximum value, showing that the yielding regions in the proximity of the hole become smaller for steel fiber-, steel-PP fiber-, and steel-PVA fiber-reinforced PCCVs. In addition, as seen in Table 5, the internal pressure that causes the steel liner to yield grows from 0.95 MPa to 0.964 MPa, 0.960 MPa, 0.963 MPa, 1.057 MPa, 1.036 MPa, and 1.019 MPa for the basalt-, carbon-, PP-, steel-, steel-PP-, and steel-PVA-reinforced containment, respectively, wherein steel fiber, steel-PP fiber, and steel-PVA fiber make the three greatest contributions and achieve increases of 11.3%, 9.0%, and 7.3% compared to the conventional structure. Predictably, the ability of steel liner to prevent the leakage of radioactive materials is further improved by considering the steel-, steel-PP-, and steel-PVA-reinforced containment.

### 4.3. Pressure Performance of Prestressing Tendons

The limit state of the structure is defined as all prestressed tendons reaching their ultimate strength. Due to the characteristics of fiber-reinforced concrete in preventing structural deformation, we can see from the curves in Figure 16 that the prestressing tendons are less stretched and the principal stress is further decreased for the tendons, particularly for steel-, steel-PP-, and steel-PVA-reinforced containments. Specifically, compared with the PCCV in service, with the addition of steel, steel-PP, and steel-PVA fibers, the safety margin of the containment is improved by 12.5%, 10.3%, and 10.4%, respectively. Further observing the failure contour in Figure 16, the ultimate internal pressure results in a similar failure state for the basalt fiber-, carbon fiber-, and PP fiber-reinforced containments. Nevertheless, for the steel fiber- and steel-PP fiber-reinforced containments, a smaller part of the “U” type prestressing tendons at the dome is caused to fail as the internal pressure reaches its maximum value. Moreover, the “U” type prestressing tendons under the equipment hatch still remain intact, apart from the smaller number of “U” type prestressing tendons that failed, particularly for the PCCV with the addition of steel-PVA fibers.

### 4.4. Evolution Rule of Cracks and Principal Tensile Strain in Concrete

The evolution rule of cracks and the maximum principal tensile strain in the concrete for the different types of containment are shown in Figure 17 and Table 6. As the pressure is lower than 0.4 MPa, there are no cracks in the structure, on account of the maximum principal tensile strain falling well below the crack strain for any type of fiber-reinforced concrete. For the structure subjected to pressure of more than 0.8 MPa, the principal tensile strain for the conventional, basalt fiber-, carbon fiber-, and pp fiber-reinforced containments tend to grow rapidly. On the contrary, the for steel fiber-, steel-PP fiber-, and steel-PVA fiber-reinforced PCCVs, the tensile strain maintains steady at an internal pressure of 1.0 MPa. As indicated in Table 6, the evolution rule of cracks for the PCCVs with fiber retrofitting is similar to that of the conventional concrete containment, while the development speed of cracks is effectively inhibited by introducing the steel fiber, steel-PP fiber, and steel-PVA fiber reinforcements. No obvious cracks develop in the steel fiber-, steel-PP fiber-, and steel-PVA fiber-reinforced containments according to the crack distribution of 0.8 MPa in Table 6, which will contribute to postponing the occurrence time of cracks. When the internal pressure is increased to 1.1 MPa, very minor cracks will occur in the vicinity of the equipment hole, and at the same time, oblique cracks develop for these three types of containment. It can be mentioned that no evident cracks appear at the bottom of the cylinder at 1.1 MPa, reflecting the uniqueness of the failure mode. Compared to the conventional PCCV, only some smaller cracks in the horizontal direction appear on the two sides of the hole, in the area below the dome and cylinder at a 1.4 MPa internal pressure. Upon raising the internal pressure up to the maximum value, severe cracks are caused in both the semispherical dome and the upright cylinder, leading to the containment being destroyed. In Table 5, it is not difficult to see that the reliable performance of the structure in resisting and staving off the occurrence of concrete cracks can be increased to varying degrees through the practice of considering any fiber reinforcement. Taking the effect of basalt, carbon, PP, steel, steel-PP, and steel-PVA reinforcement into consideration, the crack-resistance of the PCCV is increased by 15.2%, 15.7%, 20.0%, 28.0%, 22.7%, and 14.9%, respectively.

### 4.5. Analysis of the Failure Mechanism

The failure mechanism of FRC containment under internal pressure is distinctive. Due to the higher peak tensile strength of the FRC, the crack occurrence time is effectively postponed, and thus, the crack-resistant capacity of the structure can be increased. As for the FRC possessing greater post-peak tensile strength (i.e., steel fiber, steel-PP fiber, and steel-PVA fiber), the speed of the yielding evolution of the steel liner is effectively delayed, and thus, the functional failure capacity provided by the steel liner can be enhanced. Meanwhile, the larger post-peak tensile strength can make FRC continually share the increasing pressure, which results in evident enhancement of the ultimate internal pressure capacity and smaller failure regions for prestressing tendons under internal pressure. Regarding the steel-PVA fiber, the ultimate internal pressure results in the “U” type prestressing tendons at the only top of the dome being destroyed, whereas for conventional concrete, not only are the prestressing tendons for the entire dome damaged under ultimate internal pressure, but so are the prestressing tendons around the equipment hatch. It is of importance to make the failure regions controlled under the ultimate internal pressure via the appropriate use of FRC.

## 5. Conclusions

The purpose of this investigation is to compare the mechanical behavior of conventional and fiber-reinforced PCCVs and accurately analyze and identify the failure mechanism of these structures under internal pressures that are caused by multiple possible accidents. Six types of fiber, including basalt fiber, carbon fiber, PP fiber, steel fiber, steel-PP hybrid fiber, and steel-PVA hybrid fiber, are chosen to create fiber-reinforced PCCVs, and the applicability of the FRC is scientifically substantiated, as well. The conclusions are summarized as follows:The failure mechanism for the fiber-reinforced containment is different from that of the containment without fiber reinforcement. Due to the higher peak tensile strength of the FRC, the crack occurrence time can be effectively retarded, and thus, the crack-resistant capacity of the structure can be increased. As for the FRC having greater post-peak tensile strength, the speed of the yielding evolution of the steel liner is effectively delayed, and thus, the functional failure capacity provided by the steel liner can be enhanced. Meanwhile, larger post-peak tensile strength can result in evident enhancement of the ultimate internal pressure capacity and smaller failure regions for prestressing tendons under internal pressure. It is demonstrated that the failure regions can be controlled under the ultimate internal pressure via the appropriate use of FRC.The results of crack evolution show that adding different fibers into concrete can effectively delay the occurrence of concrete cracks and inhibit the development speed of cracks to varying degrees. The internal pressure of the structure to resist concrete cracks is increased within that interval [14.9%, 28.0%], with the steel fiber achieving the largest contribution of 28%.The yielding pressure of the steel liner can be elevated by 11.3%, 9.0%, and 7.3% for steel-, steel-PP-, and steel-PVA-reinforcement, respectively. It is very beneficial to reduce the possibility of leakage occurrence in case of emergency through using these three types of fiber reinforcement. At the stage of ultimate pressure, the steel lining becomes plastic in the head and at the bottom of the equipment hatch, and plasticity regions tend to develop around the equipment hatch hole.The ultimate bearing capacity of the structure can be increased by about 12.5%, 10.3%, and 10.4%, respectively, with the addition of steel, steel-PP, and steel-PVA. As mentioned above, it is recommended that steel, steel-PP, and steel-PVA reinforcement be considered, as ultimate pressure capacity is the primary goal. However, due to the complex service environments of the containment, the mechanical properties of the steel fiber may be corroded in service and the long-term performance of the containment can thus be greatly degraded. In this case, the hybrid fiber in terms of steel-PP and steel-PVA may be a better alternative and needs further investigation.

## Figures and Tables

**Figure 1 materials-16-01463-f001:**
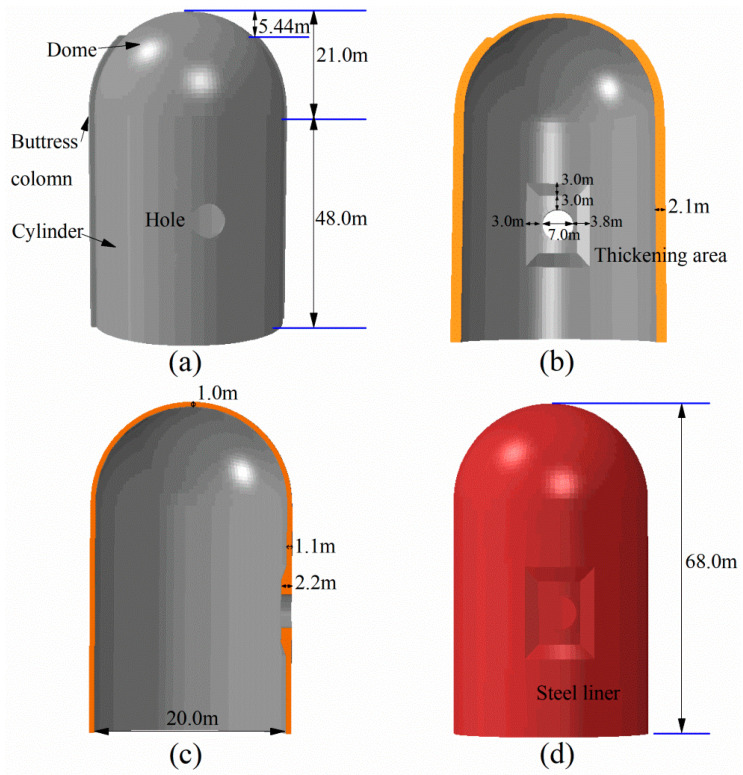
Schematic of the containment and steel liner: (**a**) 3D view of the whole structure; (**b**) x-profile; (**c**) y-profile; (**d**) 3D view of steel liner.

**Figure 2 materials-16-01463-f002:**
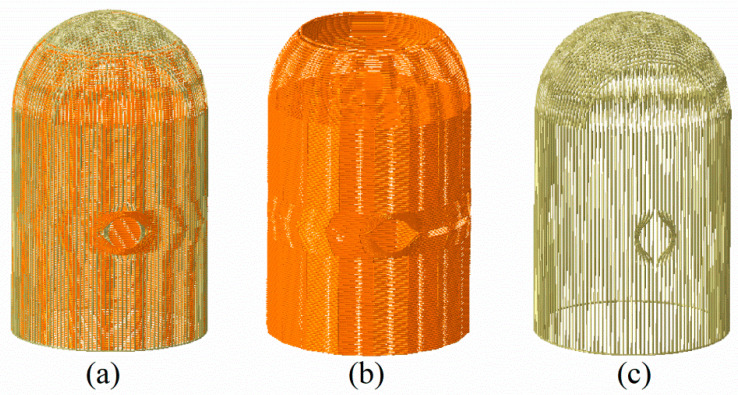
Highlighting tendon layout: (**a**) total of tendons, (**b**) hoop, (**c**) inverted “U” tendons.

**Figure 3 materials-16-01463-f003:**
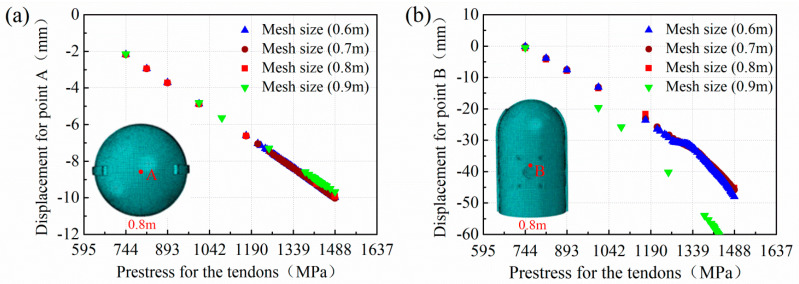
Mesh sensitivity analysis: point A (**a**); point B (**b**).

**Figure 4 materials-16-01463-f004:**
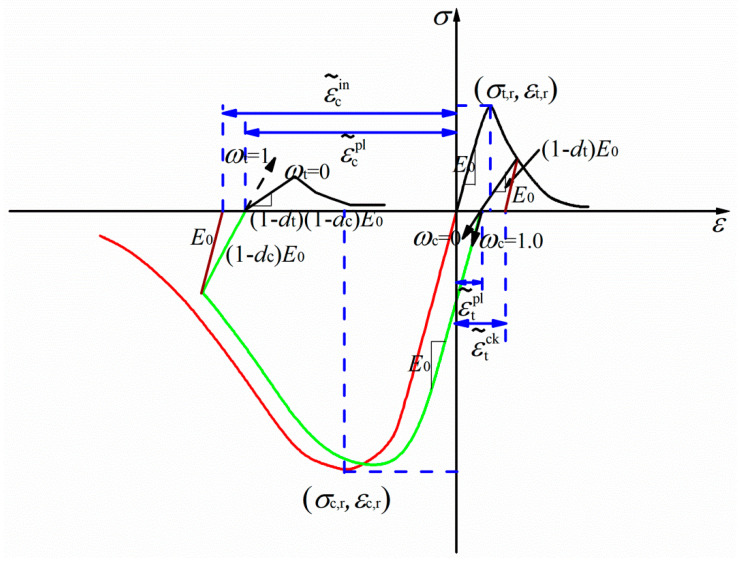
Uniaxial stress–strain curve of concrete. (Note: The curves connected by red and black colours represent the uniaxial stress–strain curves without degradation, while the curves connected by green and black colours represent the damage constitutive model of concrete.)

**Figure 5 materials-16-01463-f005:**
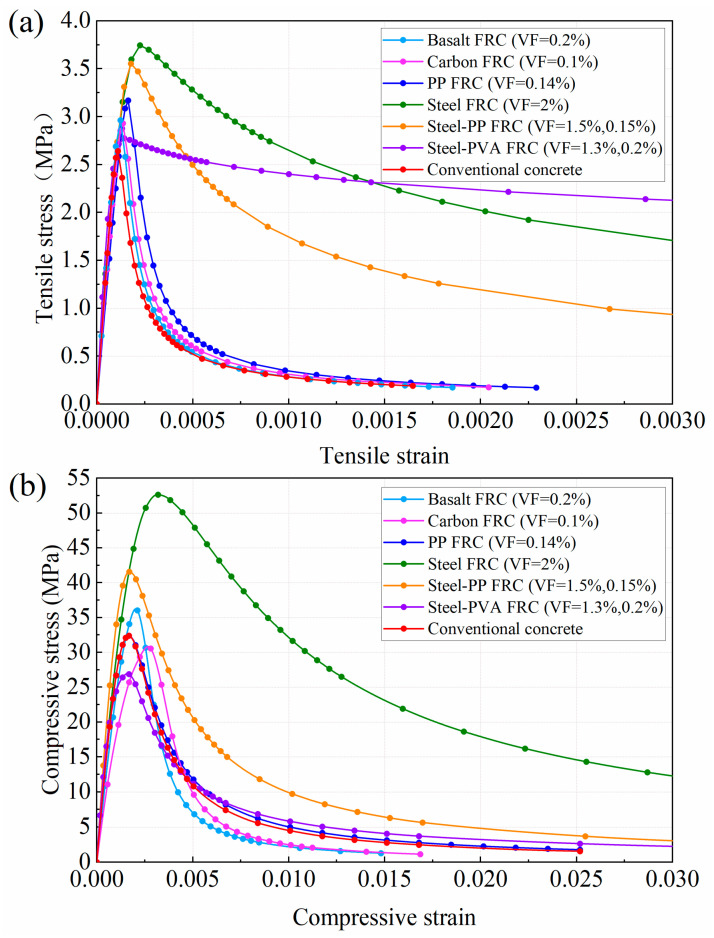
The stress–strain relationship of FRC: (**a**) Uniaxial compressive stress-strain curves; (**b**) Uniaxial tensile stress-strain curves.

**Figure 6 materials-16-01463-f006:**
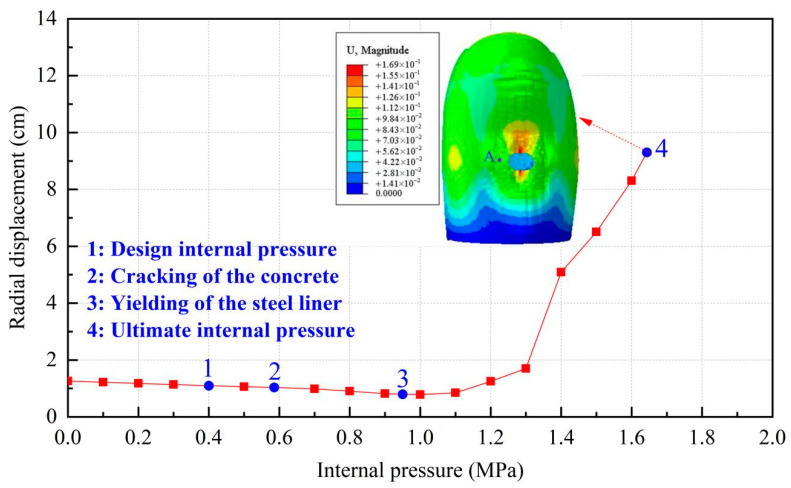
Radial displacement of the conventional containment (point A).

**Figure 7 materials-16-01463-f007:**
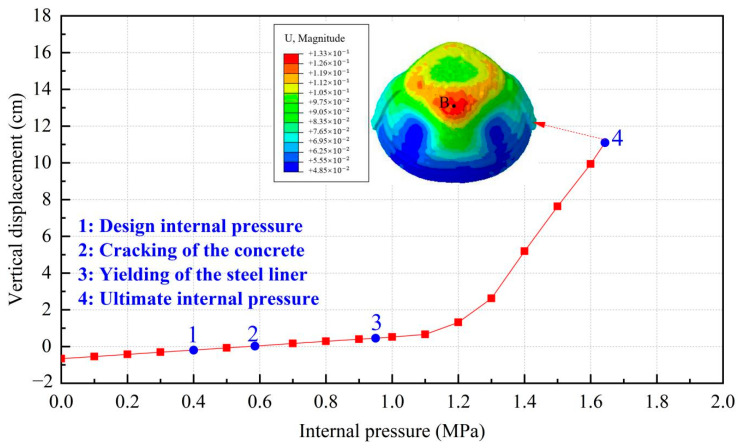
Vertical displacement of the conventional containment (point B).

**Figure 8 materials-16-01463-f008:**
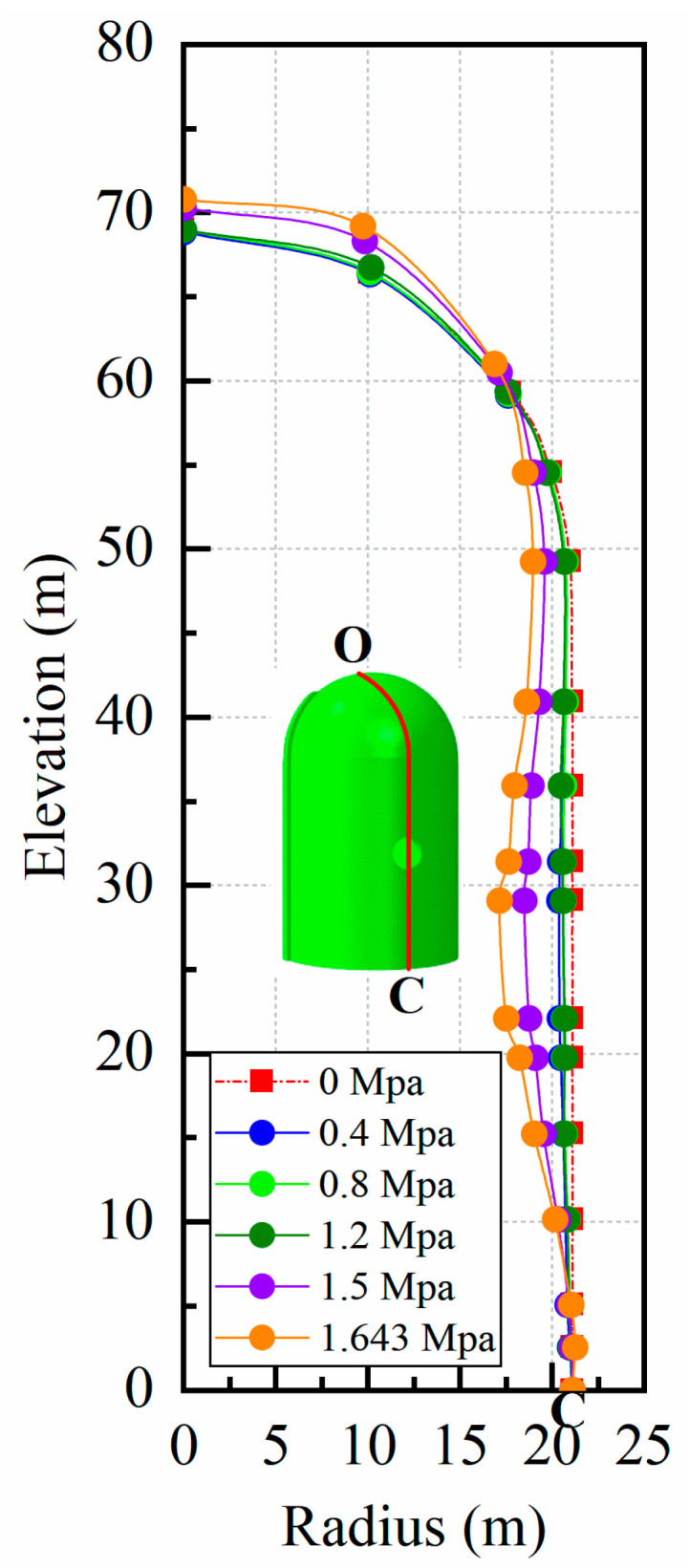
Displacement contour of the conventional containment on a large scale of 1:25 (Note: point O is at the top of the dome and point C is at the bottom of the containment).

**Figure 9 materials-16-01463-f009:**
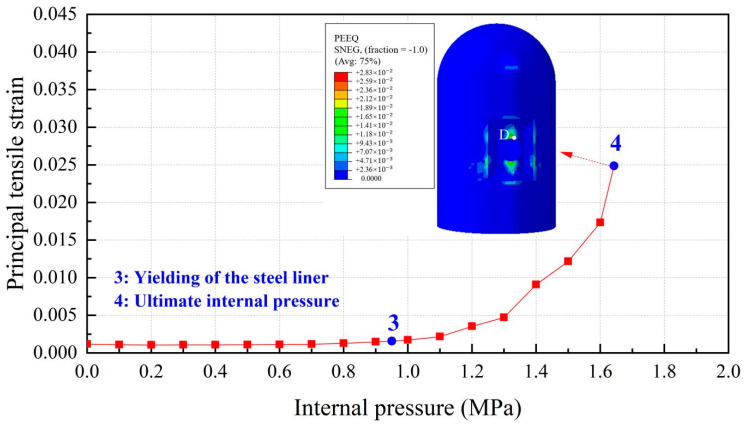
Principal tensile strain of the steel liner (Note: point D is at the apex of the hole and the red squares represent the principal tensile strain of steel liner).

**Figure 10 materials-16-01463-f010:**
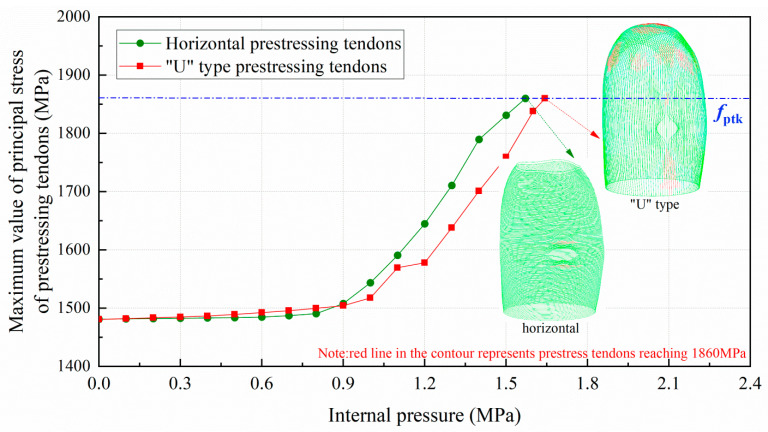
Maximum value of principal stress of prestressing tendons.

**Figure 11 materials-16-01463-f011:**
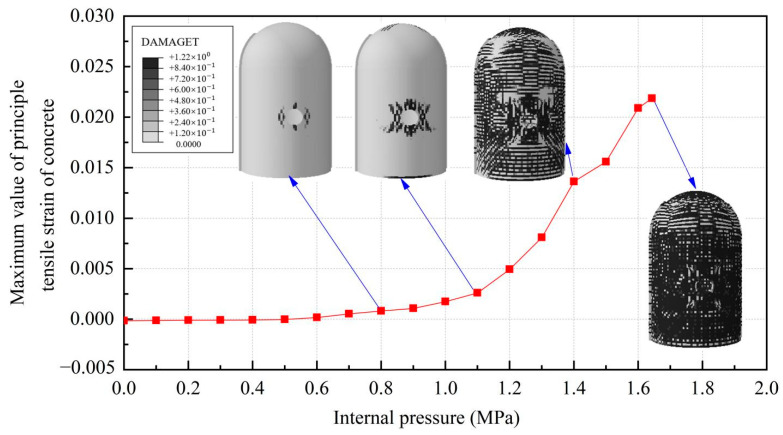
The evolution of cracks and principal tensile strain in concrete.

**Figure 12 materials-16-01463-f012:**
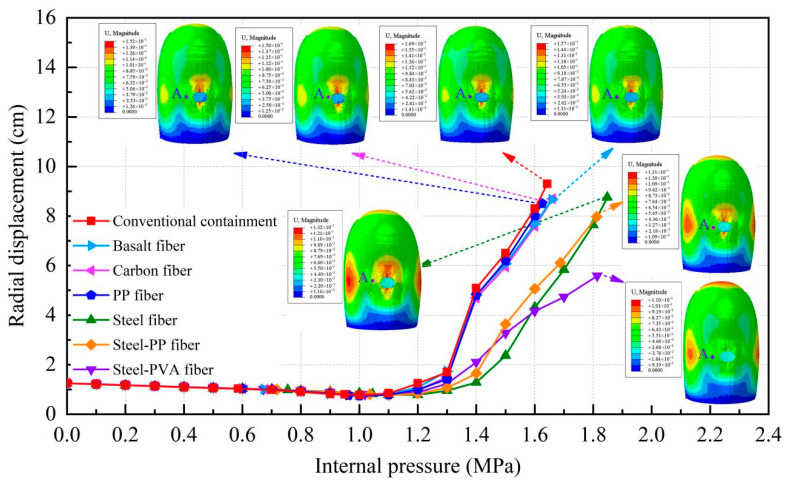
Radial displacement curve of point A.

**Figure 13 materials-16-01463-f013:**
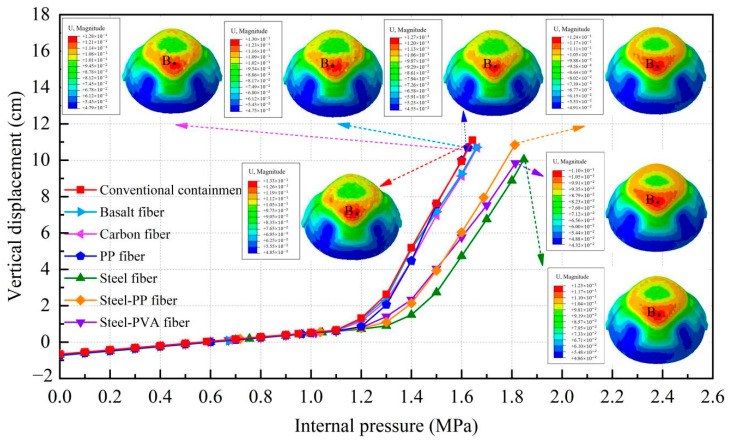
Vertical displacement curve of point B.

**Figure 14 materials-16-01463-f014:**
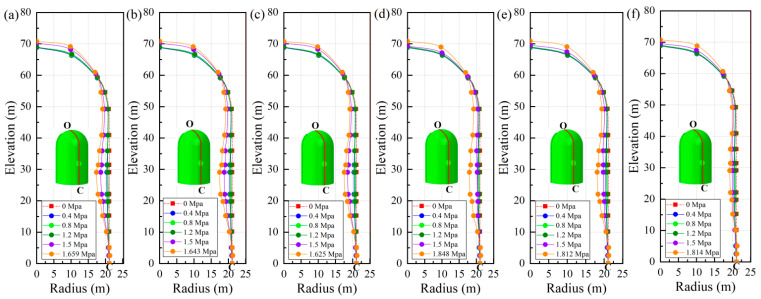
Deformation profile of the fiber-reinforced containment on a large scale of 1:25 (Note: point O is at the top of the dome and point C is at the bottom of the containment): (**a**) basalt fiber; (**b**) carbon fiber; (**c**) PP fiber; (**d**) steel fiber; (**e**) steel-PP fiber; (**f**) steel-PVA fiber.

**Figure 15 materials-16-01463-f015:**
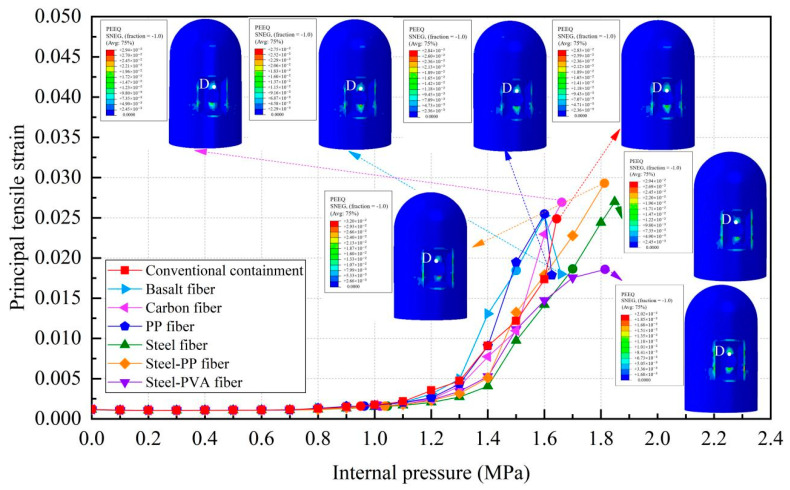
Principal tensile strain of the steel liner (Note: point D is at the apex of the hole).

**Figure 16 materials-16-01463-f016:**
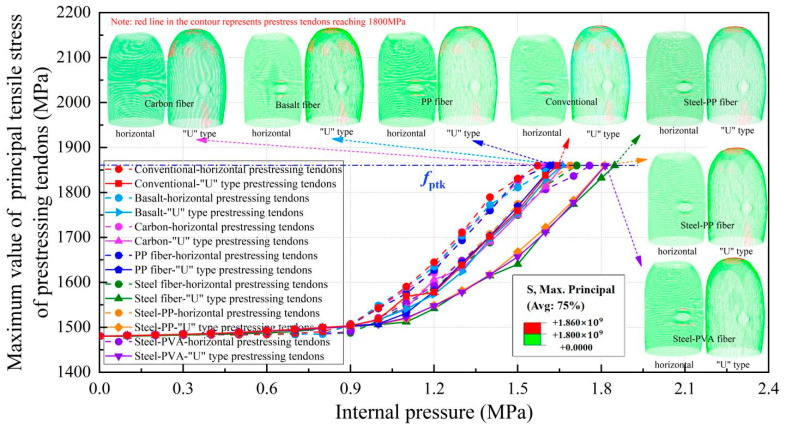
The variations in prestressed tendons’ maximum principal stress.

**Figure 17 materials-16-01463-f017:**
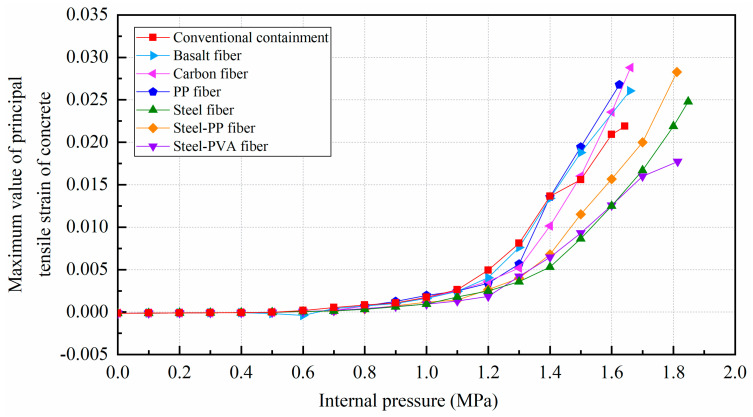
The maximum value of principal tensile strain of different concretes.

**Table 1 materials-16-01463-t001:** Material parameters of C50 concrete.

Index	Value
Density (kg/m^3^)	2500
Poisson’s ratio	0.2
Elastic modulus *E*_0_ (×10^4^ MPa)	3.45
Peak intensity—compressive (MPa)	32.4
Peak intensity—tensile (MPa)	2.64

**Table 2 materials-16-01463-t002:** Plasticity parameters of C50 concrete.

Dilation Angle (ψ)	Eccentricity (*e*)	$\sigma_{b0}/\sigma_{c0}$	$K_{c}$	Viscosity Parameter
36°	0.1	1.16	0.667	0.005

**Table 3 materials-16-01463-t003:** Properties of selected fibers.

Fiber Type	Density (kg/m^3^)		Elastic Modulus (GPa)		Tensile Strength (MPa)		Length/Diameter		Volume Fraction (%)	
Basalt	2750		105		4256		1000		0.2	
Carbon	1780		238		3900		1000		0.1	
PP	910		6.5		>400		396		0.14	
Steel	7800		200		1345		60		2	
Steel-PP	7800 (Steel)	910 (PP)	200 (Steel)	>3.5 (PP)	1225 (Steel)	>400 (PP)	60 (Steel)	167 (PP)	1.5 (Steel)	0.15 (PP)
Steel-PVA	7800 (Steel)	1300 (PVA)	200 (Steel)	41 (PVA)	1225 (Steel)	1560 (PVA)	65 (Steel)	300 (PVA)	1.3 (Steel)	0.2 (PVA)

**Table 4 materials-16-01463-t004:** The material property parameters of the steel material.

Type	Density (kg/m^3^)	Poisson’s Ratio	Elastic Modulus *E*_0_ (GPa)	Yield Strength *f*_yk_ (MPa)	Ultimate Strength *f*_ptk_ (MPa)
Prestressing tendons	7850	0.3	200	/	1860
Reinforcing steel	7800	0.3	195	400	/
Steel liner	7800	0.3	200	320	/

**Table 5 materials-16-01463-t005:** The internal pressure between conventional and FRC containment.

Containment	Concrete Cracking (MPa)	Steel Liner Yielding (MPa)	Ultimate Internal Pressure (MPa)
Conventional	0.585	0.950	1.643
Basalt fiber	0.674	0.964	1.659
Carbon fiber	0.677	0.960	1.661
PP fiber	0.702	0.963	1.626
Steel fiber	0.749	1.057	1.848
Steel-PP fiber	0.718	1.036	1.812
Steel-PVA fiber	0.672	1.019	1.814

**Table 6 materials-16-01463-t006:** The crack evolution of concrete between the FRC containments.

Fiber Types	*d* _t_	0.8 MPa	1.1 MPa	1.4 MPa	Ultimate Pressure
Basalt fiber	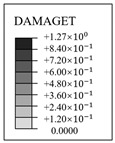	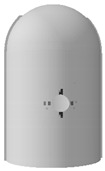	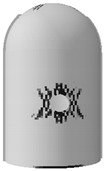	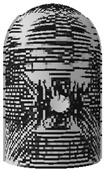	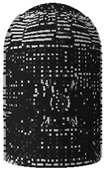
Carbon fiber	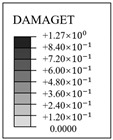	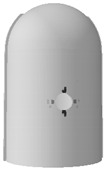	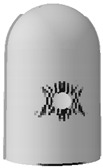	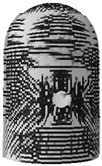	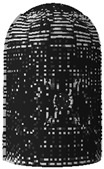
PP fiber	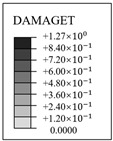	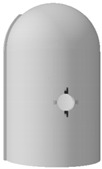	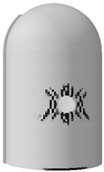	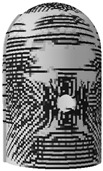	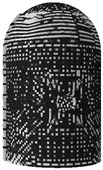
Steel fiber	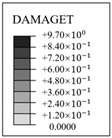	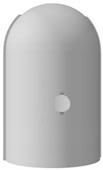	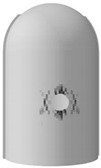	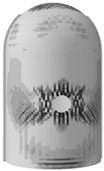	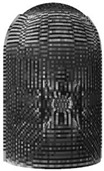
Steel-PP fiber	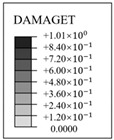	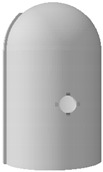	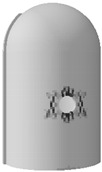	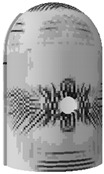	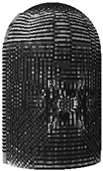
Steel-PVA fiber	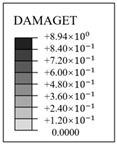	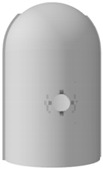	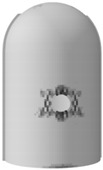	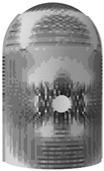	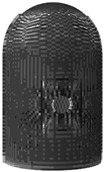

## Data Availability

All data, models, or code supporting the results of this study are available from the authors upon reasonable request.

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
