# Peer review of "Failure Mechanism of Fiber-Reinforced Prestressed Concrete Containments under Internal Pressure Considering Different Fiber Types"

_materials, 2023, doi:10.3390/ma16041463_

Round 1
Reviewer 1 Report
1. What is the main question the study addresses?
The authors simulated the behavior of a protective sarcophagus for nuclear power plants and showed the most effective types of dispersed reinforcement in the composition of concrete.
2. Do you think the topic is original or relevant in the field? Does it address a specific gap in this area?
The topic is absolutely relevant in the light of past accidents at nuclear power plants and possible repetitions of such incidents.
3. What does it add to the subject area compared to other published material?
The authors considered six types of dispersed aramature and their combinations, which is absent in previous works on this topic.
4. What specific improvements should authors consider regarding methodology? What additional controls should be considered?
Due to the fact that the protective structure is constantly exposed to ionizing radiation, which changes the characteristics of not only concrete and the steel inner shell, it makes sense to establish the degree of degradation of materials from such destructuring factors during the operation of the structure.
5. Are the conclusions consistent with the evidence and arguments presented, and do they answer the main question posed?
The conclusions are quite consistent with the tasks and questions set by the authors.
6. Are the links relevant?
Unfortunately, I can't give a qualified answer to this question because I don't have full access to the full texts of the links.
7. Please include any additional comments on tables and figures.
I cited minor comments on the tables in the review of the article, as for the figures, they are oversaturated with graphic information and are quite difficult to perceive when analyzing them.
There was a discrepancy between the dimensions of the height of the structure in Figure 1a - 69 m, in Figure 1d - 68 m;
The figure caption in figure 3 in the word mesh must begin with a capital letter;
In table 1, it is necessary to correct MPa in lines 5 and 6;
In table 3, the last two rows with indicators should be aligned with the columns
Reviewer 2 Report
Dear Authors,
The paper Failure mechanism of fiber-reinforced prestressed concrete containments under internal pressure considering different fiber types by Zhi Zheng, Ye Sun, Xiaolan Pan, Lianpeng Zhang is well suited for journal Materials. The authors of this article analyzed the results of the present studies on the significance of using concrete fiber reinforcement in prestressed concrete containment vessels.
The paper is interesting and scientifically valuable. The paper contains parts in good order. The Abstract is really the summary of the article. The length of the abstract is good. Keywords - no comments. The length of the abstract is short, but enough to put a lot of information, summarize the article. Introduction present background of analyzed problem, literature review shows important achievements of earlier articles. It should be emphasized that the authors in the last paragraphs of the introduction specified their own contribution presented in the article.
The article has been prepared carefully. The discussed topic has been thoroughly analyzed and the conclusions seem complete and correct.
The article was written enough well in English, is understandable for a reviewer, a person who does not speak English as a mother tongue.
General comments:
1. The authors consider the influence of pressure, what about temperature? Probably soon after the failure it will be high in steel liner and later also in concrete.
2. Have the authors considered the influence of the temperature inside at the time of the reactor accident on the PP fibers and the preservation of tightness?
3. Have the authors considered the aspects of FRC degradation over time, can the assumptions be trusted?
4. Unfortunately, the readability of graphs with images is not sufficient. Charts should be corrected in terms of readability, enlarged, text adjusted.
5. Fig. 6 and fig. 12 – request for comment why the displacement in the range of 0.4-1.0 MPa has decreasing values despite the increase in pressure.
6. The bibliography refers to 45 articles and other publications. However, the reviewer find only one reference to journal Materials in the bibliography.
Detailed comments:
- page 15, line 7 – ”1.57Mpa for horizontal tendons and 1.64Mpa” – should be “MPa”.
- Fig. 8 – which means "scale: 25" in the caption.
Round 2
Reviewer 2 Report
Dear Authors,
I have carefully analyzed the comments of all reviewers and all the responses of the authors. In my opinion, the authors were able to accurately explain all uncertainties and answer all questions. The article has been corrected.